# Becoming metrics literate: An analysis of brief videos that teach about the h-index

**Lauren A. Maggio** [1]*, **Alyssa Jeffrey**[2,3], **Stefanie Haustein**[2,3], **Anita Samuel**[1]

**1** Uniformed Services University of the Health Sciences, Bethesda, Maryland, United States of America,
**2** University of Ottawa, School of Information Studies, Ottawa, Ontario, Canada, **3** Scholarly Communications Lab in Ottawa and Vancouver, Vancouver, Canada

* lauren.maggio@usuhs.edu

## Abstract

### Introduction

Academia uses scholarly metrics, such as the h-index, to make hiring, promotion, and funding decisions. These high-stakes decisions require that those using scholarly metrics be able to recognize, interpret, critically assess and effectively and ethically use them. This study aimed to characterize educational videos about the h-index to understand available resources and provide recommendations for future educational initiatives.

### Methods

The authors analyzed videos on the h-index posted to YouTube. Videos were identified by searching YouTube and were screened by two authors. To code the videos the authors created a coding sheet, which assessed content and presentation style with a focus on the videos' educational quality based on Cognitive Load Theory. Two authors coded each video independently with discrepancies resolved by group consensus.

### Results

Thirty-one videos met inclusion criteria. Twenty-one videos (68%) were screencasts and seven used a "talking head" approach. Twenty-six videos defined the h-index (83%) and provided examples of how to calculate and find it. The importance of the h-index in high-stakes decisions was raised in 14 (45%) videos. Sixteen videos (52%) described caveats about using the h-index, with potential disadvantages to early researchers the most prevalent (n = 7; 23%). All videos incorporated various educational approaches with potential impact on viewer cognitive load. A minority of videos (n = 10; 32%) displayed professional production quality.

### Discussion

The videos featured content with potential to enhance viewers' metrics literacies such that many defined the h-index and described its calculation, providing viewers with skills to recognize and interpret the metric. However, less than half described the h-index as an author quality indicator, which has been contested, and caveats about h-index use were

**Data Availability Statement:** All files are available from Zenodo (doi:10.5281/zenodo.5885191).

**Funding:** SH and Aj were supported by funds from the Social Sciences and Humanities Research Council of Canada (SSHRC) Insight Grant #435-

2021-0108 "Metrics Literacies: Improving the understanding and appropriate use of scholarly metrics in academia" (https://www.sshrc-crsh.gc. ca/) The funders had no role in study design, data collection and analysis, decision to publish, or preparation of the manuscript.

**Competing interests:** The authors have declared that no competing interests exist.

inconsistently presented, suggesting room for improvement. While most videos integrated practices to facilitate balancing viewers' cognitive load, few (32%) were of professional production quality. Some videos missed opportunities to adopt particular practices that could benefit learning.

## Introduction

In academia, citation and publication metrics, such as the journal impact factor and h-index, are used to make critical decisions about individuals, including those decisions that govern hiring, promotion, retention and funding. The importance of quantitative metrics has created a pressure to publish, leading to a range of adverse effects and scientific misconduct, including duplication of publications, gratuitous self-citations, 'citation cartels,' self-plagiarism and so-called 'salami publishing' [1–4]. The quantification and oversimplification of research output and impact is harming scholarly communities in all disciplines [5]. These high stakes create an imperative that academics, including administrators, have a robust understanding of how metrics are derived as well as basic knowledge about their strengths and weaknesses. In essence, academic decision makers must be metrics literate. Unfortunately, this does not seem to be the case. Multiple studies document metrics misuse [6–12]. In 2020, Ioannidis and Boyack advocated for training in metrics, pointing out that "the lack of training in what they [metrics] mean and what they can and cannot tell us may be the greatest threat associated with them" [13].

M*etrics literacies*, which are related to the concept of being metrics wise [14], are an integrated set of competencies, dispositions and knowledge that empower individuals to recognize, interpret, critically assess and effectively and ethically use scholarly metrics. We argue that metrics literacies are essential among academics and administrators and that short videos are most suitable to educate them [14–16]. Thus, in this manuscript we attempt to characterize, one educational approach, the use of short online videos, as an approach for training in order to provide recommendations for future educational initiatives.

Scholarly metrics, or more specifically bibliometric indicators, are quantitative statistical measures based on publication and citation counts. Used carefully and as a complement to qualitative evaluation approaches such as peer review, they can, to a certain extent, inform about research productivity, collaboration and impact of individual authors, teams, universities and even countries [17,18]. Unfortunately, the application of bibliometric indicators can be largely characterized by inappropriate use of simplistic and limited indicators in the context of hiring, promotion and funding. The most popular metrics are not those carefully constructed by bibliometric experts, but those that are easily available. The journal impact factor and the h-index are the most popular bibliometric indicators, which despite their known flaws and limitations, are still heavily used and can have significant career implications.

### About the h-index

The h-index was created in 2005 by physicist Jorge Hirsch as a "simple and useful way to characterize the scientific output of a researcher" [19], trying to combine the two dimensions of scientific productivity (i.e., number of publications) and impact (i.e., number of citations). Mathematically, it is a simple index defined as the largest number of h papers with at least $h$ number of citations, meaning that a researcher with an h-index of 15 has published at least 15 papers which have received at least 15 citations each. Hirsch [19] claimed that the h-index is

an "estimate of the importance, significance, and broad impact of a scientist's cumulative research contributions", but due to the arbitrary, methodologically questionable [20], combination of the publications and citations, it lacks a clear concept.

In addition, the h-index has a range of flawed properties which lead to inconsistent results [18,21,22]. The most striking inconsistency appears in the context of absolute performance improvement: if two authors publish exactly the same number of additional publications with exactly the same amount of citations, the ranking of these authors relative to each other should remain the same. However, the h-index does not behave that way [22]. It produces inconsistent results insofar as it could either increase or stay the same for both authors but could also, counterintuitively, increase for one author, while it stays the same for the other [18]. Due to these inconsistent properties it "cannot be considered an appropriate indicator" [22].

Created to assess a scientist's career, the h-index is a time- and size-dependent indicator, meaning that it depends on the duration of each scientist's career and their total number of publications [18,23,24]. By definition it therefore disadvantages early career researchers [25,26]. Similar to the journal impact factor, the h-index also lacks field normalization [26,27], which means that it does not account for differences in publication and citation practices across disciplines [28]. Therefore, the h-index should not be used to compare researchers from different fields [19,26,28]. Additionally like most other bibliometric indicators, the h-index can vary depending on which database is used based on publications and citations indexed and how the database identifies citations [20,26]. For example, an author's h-index reported in Google Scholar is almost always larger than that reported by the Web of Science because Google Scholar covers more publications than Web of Science.

Despite its range of serious shortcomings, the h-index frequently informs the granting of tenure and promotion, as well as allocating academic prizes and grants [10,25,29]. While some universities ask applicants to provide their h-indices, others set h-index thresholds to short-list candidates for the appointment of different academic ranks [6,11,30,31]. Even if not used officially, individual evaluators might consult the h-index behind the scenes [9]. When acting as evaluators, researchers, often under time pressure, feel that they need an objective measure to compare applicants [10]. Thus, there is a need for training and related educational resources to ensure that this and other scholarly metrics is well understood and utilized appropriately [13,15,16].

## Educational approaches

There have been some attempts to address the lack of metrics literacies in the wider academic community. The *San Francisco Declaration on Research Assessment (DORA)* includes recommendations for improving the evaluation of research output by refraining from using the journal impact factor to evaluate individuals [32], while the *Leiden Manifesto* [33] lists ten principles to guide the use of bibliometrics in research evaluation. The *Metric Tide* [34] introduces a framework for responsible metrics. Other efforts include books and guides, such as *Measuring Research* published in Oxford University Press's What Everyone Needs to Know series [17], which provides an overview of scholarly metrics in an accessible manner. Similarly, *Becoming Metric-Wise*: *A Bibliometric Guide for Researchers* [18] seeks to increase the knowledge about bibliometric methods and indicators among researchers. The *Metrics Toolkit*, takes a similar step towards communicating scholarly metrics to a general academic audience by providing "evidence-based information about research metrics across disciplines" [35]. These initiatives represent a valuable first step toward educating the broader academic community on the appropriate use of scholarly metrics. However, they are all published as text-heavy articles, monographs, or websites, which require hours of reading—one of the slowest and most

laborious means for acquiring and retaining knowledge [36]. Videos, on the other hand, are perceived by learners as more engaging [37] and easily accessible [38]. Thus, we have decided to focus our energies on video as a more efficient and effective online education format.

Videos are effective educational tools [39–41] that encourage a multi-sensory learning experience [38]. Videos have long been used for educational purposes [42] and their use has been demonstrated to increase student motivation by approximately 70% [43]. Videos are effective in conveying complex information. By *showing* educational content instead of simply *telling* it [44], videos can spark learner interest [45–48], as well as improve their comprehension and retention of the material [48,49]. These benefits may be particularly valuable for teaching complex topics (such as scholarly metrics), as they can provide guidance and clarity and increase learners' engagement [48,49]. For example, video lectures and podcasts can raise learners' interaction with academic content, allow them to set their own pace of instruction and improve their learning experience [50–53].

Video use for educational purposes has exploded since the arrival of YouTube, a freely-available, web-based video platform. A recent survey found that 51% of adults in the United States use YouTube to learn [54]. In today's educational landscape, learners are increasingly spending time on mobile devices, creating a demand for content that is mobile-accessible and tasks that can be performed on-the-go [52]. As a free platform, YouTube removes barriers to entry; anyone can be a content creator or consumer. With more than 2 billion active users and approximately 500 minutes of content uploaded every minute [55] YouTube provides accessible content on various platforms including mobile devices, which encourages learning on-the-go and provides just-in-time learning opportunities.

However, little is known about how the h-index and other bibliometric indicators have been covered on this platform, raising questions about the characteristics of these videos, including what content is covered and educational techniques utilized. Thus, in this study, we aimed to characterize the freely available YouTube videos focused on the h-index in order to understand the state of the available resources and to provide practitioners with practical findings to optimize the creation of future videos.

## Methods

We conducted an analysis of publicly-accessible videos on the h-index posted to YouTube. As this study did not involve humans, we did not submit this research to an ethics board.

On August 26, 2021 we identified 274 videos by searching YouTube via Google.ca ("h index" OR "hirsch index" OR "h-index" site:youtube.com). The three results pages were downloaded as HTML for future access. These 274 videos became our initial data set. Each of the 274 videos was independently viewed by at least two reviewers and considered in relation to our inclusion and exclusion criteria. We included videos about the h-index. We considered a video to be about the h-index if the presenters described the metric, such as providing a description of how the h-index is calculated (e.g., A scientist has index h if h of their Np papers have at least h citations each and the other (Np–h) papers have ≤h citations each.). We excluded videos that did not describe the metric, but instead either just mentioned it as a generic citation metric or focused on tasks like how to write code to calculate an h-index. We also excluded results that were only a YouTube playlist, but did not contain content about the metric. For feasibility, we excluded videos not presented in English. Based on research by Guo et al. [56], who found that median engagement time with videos is about six minutes and viewers watch less than halfway through videos that are longer than nine minutes, we also excluded videos over 10 minutes in duration. We did not limit the date range of our search, however, the h-index was first proposed in 2005 [19], so all retrieved videos were published after 2005.

To determine inclusion, all 274 videos were independently viewed by at least two authors. All discrepancies were resolved by group consensus.

To capture video characteristics, we collaboratively created a codebook based on our experience and training as educators as well as the literature on video education [52,57–60]. To determine the videos' educational quality, we anchored our evaluation on the related work by Young and colleagues [61]. Young et al. [61] considered the efficacy of educational videos for adult learners based on how the videos increased or decreased a viewer's cognitive load thus influencing their learning. Cognitive load is described as the amount of information that an individual can hold at one time in their working memory, which is a limited resource [62]. If a learner's working memory is overloaded, then learning is negatively impacted [63]. Researchers have proposed that educators use Cognitive Load Theory (CLT), which is an instructional theory, to guide educational design in order to optimize learners' cognitive load [64]. CLT, which was introduced by Sweller in 1988, posits that there are three types of cognitive load: intrinsic, extraneous, and germane load that must be accounted for and balanced when designing instruction [62]. In our coding, we identified elements within the videos with the potential to impact each type of cognitive load. To make judgements about the production quality of the videos, which we classified broadly as amateur or professional, we relied upon video characteristics as described by industry [65]. The codebook and data resulting from this study and related codes are freely available on Zenodo [66]. The coding tool was operationalized in Google Sheets.

AJ, AS, and LM conducted three pilot rounds to test the efficacy of the coding sheet on three separate sets of videos. Following each pilot round, the coders discussed the fitness of the coding sheet for the task and made any necessary updates. After finalizing the coding sheet, each video was coded in duplicate by two authors with each coder completing their task independently. Discrepancies between two coders were resolved by group consensus by all authors, with SH acting as a tiebreaker, as needed.

As noted, we were provided access to 274 videos, however, as our aim was to understand the characteristics of educational videos about the h-index instead of characterizing all available videos on this topic, we did not aim to comprehensively identify each video about the h-index. Instead, we worked towards data sufficiency, which is the point at which we could derive a clear and coherent understanding of key characteristics of the videos and could identify no additional nuances or insights [67]. After each round of coding, we discussed via video conference if we felt that we had reached data sufficiency. We felt that we reached data sufficiency at 24 videos. However, we reviewed an additional seven videos for certainty, which did not yield any new insights Our final dataset therefore contained 31 videos. Data from our coding was compiled in Google Sheets and described using descriptive statistics.

## Results

We analyzed 31 videos and report their characteristics, content, and factors that have potential to impact a viewer's cognitive load.

### Video characteristics

Videos were posted to YouTube between 2011–2021 with a majority (81%) posted between 2019–2021. On average, videos were 5.12 minutes in duration (range = 0.47–9.59; SD = 2.89). Comments had been posted to 20 of the videos with the caveat that some of the videos did not allow for comments. The number of views the videos had received ranged from 10 to 20,503 (AVG = 2749; SD = 4618). The most viewed video (20,503 views) was posted by Curtin Library in 2013 called *the H-Index Explained* (Video 2). See Table 1 for a listing of all videos and key characteristics.

**Table 1. Characteristics of YouTube videos on the h-index (n = 31).**

| Video # | Unique Identifier | Title | Video Duration | Year posted | # views | Video type | Video topic | Video quality |
|---|---|---|---|---|---|---|---|---|
| 1 | 1 | What is the h-index? | 3:40 | 2018 | 13069 | talking head | h-index | Amateur |
| 2 | 2 | H-Index Explained | 6:14 | 2013 | 20521 | screencast | h-index | Professional |
| 3 | 3 | EP 05: h-index | 2-MIN METRICS SERIES | 2:40 | 2020 | 410 | animation | h-index | Professional |
| 4 | 4 | What is an H-index? | How To Research | 2:12 | 2021 | 49 | talking head; screencast; animation | h-index | Professional |
| 5 | 5 | How to Find out the H-Index of an Author | Author-Level Metrics | Journal Publications & Citations | 1:39 | 2021 | 56 | screencast | h-index; other metrics | Professional |
| 6 | 7 | What is H INDEX | 8:03 | 2020 | 487 | screencast | h-index; other metrics | Amateur |
| 7 | 8 | h Index | 9:26 | 2020 | 6107 | screencast | h-index; other metrics | Amateur |
| 8 | 10 | What is h-index | 2:23 | 2021 | 27 | talking head | h-index | Professional |
| 9 | 11 | h - Index: What is h- Index? How To Calculate h-index? | 6:34 | 2020 | 11620 | screencast | h-index; other metrics | Amateur |
| 10 | 12 | H - Index in Google Scholars: Why h-index is Important for Your Research Career? | 7:26 | 2019 | 4605 | screencast | h-index; other metrics | Amateur |
| 11 | 14 | Understanding Scopus h-index - How does it increase? | 6:38 | 2020 | 1,705 | screencast | h-index | Amateur |
| 12 | 17 | What is h index and i10 index | phd | Milton Joe | 8:55 | 2020 | 3,187 | screencast | h-index; other metrics | Amateur |
| 13 | 20 | What is h-index | 5:55 | 2020 | 1,601 | talking head; screencast | h-index | Amateur |
| 14 | 22 | Citations, i10-index, h-index and m-index | 8:41 | 2020 | 441 | screencast | h-index; other metrics | Amateur |
| 15 | 23 | Scopus Tip: How to increase your H index and citation in Scopus (English Version) | 8:30 | 2020 | 1,724 | screencast | h-index | Amateur |
| 16 | 26 | Citation INDEX | How To Calculate i-10 Index and H-Index | 2:32 | 2020 | 1,888 | animation | h-index; other metrics | Amateur |
| 17 | 30 | Why the h-index cannot be compared across disciplines | 4:28 | 2020 | 291 | interview | h-index | Amateur |
| 18 | 33 | h index | how to calculate h-index | h-index of an Author | Journal | University or Country h index | 3:26 | 2021 | 689 | screencast | h-index | Professional |
| 19 | 37 | Google scholar citation| h-index and i10 index| Progress with Prof.Mahamani | 9:34 | 2020 | 483 | talking head; screencast | h-index; other metrics | Amateur |
| 20 | 40 | Hirch index (h-index) | Citation Index (Notes Included) | 3:08 | 2021 | 47 | screencast | h-index | Amateur |
| 21 | 44 | What is the h-index? in 5 Min ONLY | 8:18 | 2021 | 227 | Screencast; scribing | h-index; other metrics | Amateur |
| 22 | 47 | h-index | 1:28 | 2019 | 47 | screencast | h-index | Professional |
| 23 | 56 | Limitations of the h-index for early career researchers | 1:20 | 2011 | 5,482 | interview | h-index | Professional |
| 24 | 58 | Importance of h index - @Dr. Anand Nayyar - Learning with Chandan | 5:12 | 2020 | 157 | interview | h-index; other metrics | Amateur |
| 25 | 67 | h index and i10 index I Research Indices I Research to Publication I Dr.V.M.M.Thilak | 4:58 | 2021 | 28 | talking head | h-index; other metrics | Amateur |
| 26 | 74 | Evaluating h-Index: metric to evaluate authors rank | 9:18 | 2016 | 2,604 | screencast | h-index; other metrics | Amateur |
| 27 | 90 | H index | 0:47 | 2017 | 3,150 | screencast | h-index | Amateur |
| 28 | 101 | How To Calculate h index i10 and i20 index | 3:36 | 2020 | 202 | talking head | h-index; other metrics | Amateur |
| 29 | 103 | H index (English) | 2:37 | 2021 | 15 | animation | h-index | Professional |

*(Continued)*

**Table 1.** (Continued)

| Video # | Unique Identifier | Title | Video Duration | Year posted | # views | Video type | Video topic | Video quality |
|---------|-------------------|-------|----------------|-------------|---------|------------|-------------|---------------|
| **30** | 148 | Impact Factor of Non SCI Journals H-Index Half Life JCR IF | 2:05 | 2020 | 35 | screencast | h-index; other metrics | Amateur |
| **31** | 156 | An Introduction to Bibliometrics | 9:59 | 2015 | 4,256 | screencast | h-index; other metrics | Professional |

The majority of videos (n = 21; 68%) were presented as screencasts (i.e., digital recordings of a computer screen usually accompanied with descriptive audio). For those using screencasts, 11 videos (35%) presented voice-over presentation slides and seven videos were recordings of the presenter navigating online resources, such as a database or library website (23%). Seven videos used a "talking head" approach (n = 7; 23%) in which the pictured presenter spoke directly to the viewer. Other formats included animation (n = 4; 13%), interviews (n = 3; 10%) and scribing (e.g., overview of a hand drawing/writing) (n = 1; 3%).

Twenty-six (84%) of the videos incorporated human presence. This presence included active depictions of human presenters (e.g., a presenter writing on a whiteboard; a presenter conducting an interview) (n = 7; 23%); still images, such as an academic headshot (n = 17; 55%); or voice-over audio by the presenter (n = 26; 84%). When addressing viewers, 19 (61%) presenters used the second-person pronoun or addressed the viewer as "we".

Seventeen presenters (55%) identified themselves in the videos; however, their roles were not always explicitly stated. Roles included faculty members (n = 10; 32%) and self-described researchers (n = 4; 13%) and three videos that included names but did not have information on who they were. Two videos (6.5%) were posted by library YouTube accounts, but it was unclear if the presenters were librarians.

The production quality of the videos could broadly be classified as amateur or professional. A majority of the videos (n = 21; 68%) displayed an amateurish production quality evidenced by poor camera angles, audio quality, and lack of editing. The presenters spoke in an unscripted "off-the-cuff" style which was characterized by repetitions and filler words. For example, "Actually because h-index—just to give you a uh, uhm, historical overview of h-index -, previously, uhhh, researcher they. . ." (Video 6). When there was no audio, the slides were text heavy and difficult to read (Video 30). The videos deemed to be of professional production quality (n = 10; 32%) incorporated professional images and animations along with scripted dialogue that flowed smoothly.

## Video content

Content in the 31 included videos varied. Most videos defined the h-index (n = 26; 83%) but only 39% (n = 12) mentioned Jorge Hirsch, the inventor of the h-index. To exemplify the calculation of h-index, 22 videos (72%) utilized fictional examples. For example, one presenter stated: "an h-index of 12 would mean that out of all the publications by a group or person, 12 articles would have received at least 12 citations each" (Video 1). However, 14 presenters (45%) utilized examples of their own h-index or that of another named researcher.

Some videos featured content on how to find an h-index (n = 10; 32%). The majority of videos (n = 19; 61%) mentioned that viewers could locate the h-index in Google Scholar, while 45% mentioned Scopus (n = 14) and 39% Web of Science (n = 12). See Table 2 for a summary of the videos' content. A few videos addressed scholarly metrics broadly (e.g., impact factor, i-10 index) (n = 8; 25%) while 10% of videos specifically mentioned how someone can increase their h-index (n = 3).

**Table 2. A summary of the content of short YouTube videos on the h-index (n = 31).**

| Content | Count (%) |
|---|---|
| Defines the h-index | 26 (84) |
| Provides a fictional example of the h-index | 22 (71) |
| Mentions other impact indicators | 16 (52) |
| Provides a real example of the h-index | 14 (45) |
| Refers to Jorge Hirsch | 12 (37) |
| **Resources for locating h-index** | **22 (71)** |
| Google Scholar | 19 (61) |
| Scopus | 14 (45) |
| Web of Science | 12 (39) |
| **Describes cautions to the h-index** | **16 (52)** |
| Disciplinary differences | 12 (39) |
| Database differences | 9 (29) |
| Disadvantage to early career researchers | 7 (23) |
| Self-citation inflation | 3 (10) |
| Author order negated | 3 (10) |
| Non-English language publications | 1 (3) |

Most videos (n = 28; 90%) described the h-index as an author-level metric. However, five videos (16%) noted that the h-index could also be used as an indicator to describe journals, while three videos (10%) mentioned its applicability to groups of authors, such as researchers based in specific universities or countries. Fifteen videos (49%) described the h-index as indicative of the importance, quality or impact of an author or set of publications. For example, one presenter notes "To determine how productive and impactful a researcher is, we have something called the h-index" (Video 4). Another presenter stated: "My professor says a good researcher, his h-index should be equal to his age. . .These people are highly cited and it means they are a good researcher. My h-index is 12 and I am 34. This means I am not a good researcher" (Video 11). Of note, while we did not code videos for accuracy, this example stood out as an example of an inaccurate statement. The importance of the h-index in recruitment, tenure, and promotion decisions was raised in 45% (n = 14) of the videos. For example, the "h-index is increasingly being used in the critical assessment of faculty for tenure and promotion alongside other forms of evaluation" (Video 31). When explaining that he includes the h-index on his CV, one researcher noted: "It is a very good factor [the h-index] to impress someone" (Video 11). To this end, three videos (10%) proposed strategies for a researcher to raise their h-index. For example, one presenter described a step-by-step process of identifying and emailing researchers copies of their articles in a bid to increase citations and ultimately their h-index (Video 23).

As we noted above, the h-index has several issues which make it a problematic metric to use in research evaluation. Over half of the videos (n = 16; 52%) raised caveats about using the h-index, especially for high-stakes situations, such as hiring and promotion and tenure decisions. For example, seven videos (23%) cautioned that using the h-index could disadvantage early career researchers as they may have published fewer articles and the citations to those articles have had less time to accrue. Twelve videos (39%) warned against comparing scholars in different disciplines as disciplines can have different citation traditions. Nine videos (29%) described that the h-index value for the same author might differ, depending on the database used. Several videos mentioned that citation practices such as self-citations can inflate an h-index (n = 3; 10%) and that the h-index does not account for author order, such that an author's h-

index is influenced equally by articles in which they were a middle author as those in which they were first author. A single video (Video 1) noted that authors publishing in languages other than English were disadvantaged in the h-index calculations, which is due to non-English publications having lower citation rates and being largely excluded from citation data-bases such as Web of Science and Scopus.

## Cognitive load

We identified elements and approaches in the videos with potential to impact a viewer's cognitive load, which can have implications for their ability to learn from the videos. We organized these factors into the three types of cognitive load: extraneous, intrinsic and germane. When designing instructional strategies based on CLT, the aim is to balance these three types of cognitive load in order to ensure that a learner's limited cognitive capacity is not overwhelmed [63]. See Table 3 for a summary of CLT types in the videos.

Extraneous cognitive load is a result of how content is presented to a viewer in such a way that it requires their cognitive processing, but does not contribute to their learning (e.g., a non-productive distraction). Almost all videos (n = 30; 97%) focused their content exclusively on the topic at hand and did not include distractors, which can help reduce extraneous cognitive load. One exception was a video that began with a cake cutting ceremony to celebrate a milestone for the presenter (Video 19). This was unrelated to the topic of the h-index and took 1:38 minute at the beginning of the 9:34 minute video. This may distract the viewer in regards to thinking about how the cake is relevant to the h-index. We observed extraneous elements in 17 videos (55%) which we assume were unintentional, but may still contribute to extraneous load, such as background noise (e.g., birds chirping, construction sounds, a ticking clock) and poor audio quality (Videos 7 and 30). Six videos (19%) integrated signaling, which is the inclusion of cues to highlight important information (e.g., highlighting words, 'pay attention to. . .')

**Table 3. Cognitive load factors identified in short YouTube videos on the h-index (n = 31).**

| | Count (%) | Example |
|---|---|---|
| **Extraneous Cognitive Load** | | |
| Extraneous elements | 17 (55) | • Background noise (Video 11, 26)<br>• Poor audio quality (Video 25)<br>• Oral cues "go to next slide" (Video 19) |
| Directly addressing viewers (Personalization) | 13 (42) | • "First of all, you must log in. . ." (Video 15)<br>• "I hope you can see there may be. . ." (Video 31) |
| Signaling | 6 (19) | • Use of magnification to focus on important text (Video 3, 6)<br>• Use of animated arrow to highlight important text (Video 20) |
| **Intrinsic Cognitive Load** | | |
| Timing of content | 22 (71) | Providing sufficient time to read text on screen before proceeding to next screen |
| Segmenting | 7 (23) | • "Let's see how to determine both . . ." (Video 5)<br>• "Now we'll see the advantages and the disadvantages." (Video 20)<br>• "The other quantitative metrics used for . . ." (Video 25) |
| Additional resources | 5 (16) | • Offering assistance "Reach out to me at RiverwindsConsulting.com with your questions." (Video 1)<br>• Providing links at the end of the video for further information "for further understanding one may refer to the following link . . ." (Video 7) |
| **Germane Cognitive Load** | | |
| Signaling | 28 (90) | "In this example, you can see that. . ." (Video 29) |
| Dual channel (Modality) | 22 (71) | Animated text with voice-over explanation (Video 3,4) |
| Interactive learning elements (Active processing) | 1 (3) | Posed a question to the viewer "Now let me see if you can solve this exercise. I will give you 10 seconds." (Video 5) |

and can decrease extraneous cognitive load by focusing the viewer on what the presenter feels is important. Additionally, 13 presenters (42%) used a personalized approach by directly addressing the viewer (e.g., "You can see that he has an h-index of 53"(Video 3)), which can decrease extraneous cognitive load as it enables the viewer to immediately consider the content from their own perspective.

Intrinsic load is considered essential to learning and is associated with the inherent difficulty of a task or content, such that the more complex or difficult the content is, the higher the cognitive load is for the learner. While the inherent difficulty of a task or content cannot be changed, the burden of this type of intrinsic cognitive load can be reduced if content is broken down or segmented for the learner. We observed that in seven videos (23%) presenters clearly segmented their content by including clear transitions between the content. However, 39% (n = 12) presented content in a single continuous block, which can increase intrinsic load. The other presenters (n = 12; 39%) moved between topics without clearly delineating the segments. We also observed that 22 of the videos (71%) incorporated text and they all provided viewers adequate time to comfortably read the presented text, which can lessen intrinsic load. Additionally, five videos (16%) referenced additional resources for viewers interested in learning more, which can help decrease the load put forth in a single video.

Germane load refers to the cognitive resources needed to facilitate learning [62].

Germane load can be increased by presenting content multimodally which uses dual channels (e.g., content is simultaneously presented using an image [visual channel] and the presenter's narration [oral channel]). Most videos (71%; n = 22) used a dual channel approach by featuring words on the screen plus audio elements. In a minority of videos (6%; n = 2), the video employed a simulated/computer-generated voice to present the content (Video 5, 18) which can increase germane load [68]. Three videos (10%) did not include audio or text-to-speech conversion tools, which is suboptimal for enhancing germane load. Some of the videos (19%; n = 6) did not include images and focused only on the presenters. For example, two videos presented a researcher talking directly at the camera without any other images or text to explain the concept. (i.e., a talking head video) (Videos 1, 4). Although the use of active learning can increase germane load, only a single video took this approach by integrating a quiz that challenged viewers to calculate the h-index based on presented data (Video, 5).

## Discussion

The use of scholarly metrics, including the h-index, for making high stakes decisions, such as granting promotion and funding, requires that the individuals using them are metrics literate, such that they can recognize, interpret, critically assess and effectively and ethically use scholarly metrics. In our discussion, we first focus on how the videos presented the h-index as a concept and consider this presentation within the context of metrics literacies. Then, we discuss the videos with respect to CLT with the intention of highlighting potential best practices for the creation of future videos on the h-index and metrics education more broadly.

The YouTube videos we analyzed have some content with potential to enhance a viewer's metrics literacies. To begin, most of the videos defined the h-index and described how it is derived, which provides viewers with skills to recognize and interpret the metric at a basic level. However, nearly half of the videos state that the h-index indicates an author's importance and the impact and/or quality of their publications. Although this aligns with Hirsch's original description of the metric, researchers, including Hirsch himself, have pointed to several important caveats to temper this claim [19,22,26]. In the analyzed videos, we observed that nearly half of presenters raised at least one caveat, such as differences in how databases calculate the metric; bias against early career researchers and scholars not publishing in English;

and differences across disciplines. However, these caveats were inconsistently presented across the videos, in that some videos raised only a single concern, and 48% did not raise caveats at all. Additionally, none of the videos raised identified concerns such as gender bias, which stems from identified biases in citation practices [69] and the way in which the metric behaves inconsistently [22]. Thus, while the majority of currently available videos introduce viewers to the h-index, there is room for improvement in terms of providing comprehensive content for viewers to enable them to critically assess and effectively and ethically use this scholarly metric. For example, future video creators might consider integrating into their videos brief case studies or creating personas of scholars from a variety of fields and backgrounds to demonstrate how some of these caveats can impact individuals [70].

Several videos described the importance of a high h-index for hiring, promotion, and funding decisions and provided strategies for researchers to boost their metric. This finding aligns with recent research that found that the use of metrics in review, promotion and tenure evaluations is widely encouraged and that they are often inappropriately portrayed as measures of 'quality' or 'prestige' [71]. The importance of the h-index along with other quantitative metrics, has in this way created a pressure to publish, entailing adverse effects or even leading to scientific misconduct [1–4,72]. While these videos are accurate–the h-index is indeed used in these high-stakes decisions–this messaging misaligns with recent initiatives. For example, DORA, endorsed by over 20,000 signatories since 2012, advocates that scholarly metrics not be used as a surrogate marker of quality for a researcher's career [32]. Thus, while it is important that individuals be made aware of the current (mis-)uses of the h-index, this awareness should be balanced with a critical assessment of the indicator and its aptness to assess researchers' careers. While any quantitative metric should only be used in addition and to complement qualitative assessments such as peer review, there are bibliometric indicators (e.g., percentile ranks, field-normalized citation rates) that are more suitable to assess research productivity and impact than the h-index.

While the existence of these h-index videos is encouraging, it is important to consider how they are presented. In this analysis, we considered potential effectiveness in relation to CLT, which posits that to optimize learning materials they should be presented in such a way as to balance extraneous, intrinsic and germane cognitive load [62,64]. Across the videos, we identified pedagogical practices with the potential to impact all three cognitive load types. For example, most videos were presented using a dual channel approach, such that textual, audio, and visual components were utilized to limit germane load. Additionally, most videos were tightly focused on the h-index with limited irrelevant elements, thus avoiding extraneous load, which can lower viewers' comprehension of the content [62]. To add to the focused nature of these presentations, the majority of videos included a human presence. Specifically, incorporating a human presence, a so-called *focaliser*, can create authenticity and connection between the educator and the learner [46]. We would encourage future video creators to add this type of focaliser as this can increase focus and enable smoother transitions into new topics, making learners feel that they are guided through complex material by a relatable human.

While many of the videos integrated practices to facilitate balancing viewers' cognitive load, some videos did not take advantage of particular practices that future video makers could leverage to benefit viewers. For example, only a single video (Video 5) utilized an active learning/processing approach, which can have a positive impact on germane load [61]. In this particular video, the presenter embedded a brief quiz about halfway into the video. Future video creators should consider how they can further integrate active learning approaches, such as providing viewers opportunities to pause the video to calculate the h-index or to self-reflect on ways in which the h-index can appropriately be used in research evaluation. Additionally, future video creators could consider the integration of knowledge pre-tests at the start of a

video which can be used to activate a learner's previous knowledge and provide them a sense of what they need to learn. Video creators may also want to consider being explicit about transitions and the delineation of segments, which can lessen intrinsic cognitive load. For example, as a presenter transitions from the definition of the h-index to where to find the metric, they could include a transition image or verbally announce the transition. Video producers might also consider providing a content overview so that going into the video the viewer is aware of the upcoming segments.

## Limitations

Our study should be considered in light of its limitations. Our inclusion criteria focused on videos posted to YouTube. It is possible that had we explored alternate platforms, such as Vimeo or TikTok or had scanned library websites, we may have identified additional videos, not posted on YouTube. However, as this was not an attempt to comprehensively characterize videos on the h-index and we feel that we reached data sufficiency, we propose that this is an adequate base for future researchers to further investigate alternate platforms. We limited our sample to videos that were less than 10 minutes in duration. It is possible had we included videos of greater length that we may have uncovered data points, however, in light of current research on video duration and effectiveness [55], we feel that this design decision was warranted. We did not include videos that were not presented in English. This may limit the usefulness of our findings to researchers who do not speak English. Additionally, we did not code the videos for accuracy. Future studies might examine this aspect, while keeping in mind that it is possible that information perceived as inaccurate at the time of coding might have been accurate at the time of a video's posting. Lastly, this study focused on content provided by videos, however, there exist other multimedia resources that convey information about the h-index, such as infographics and podcasts. Future researchers should consider also examining these types of resources.

## Conclusion

Given the prolific use in high-stakes decisions affecting academic careers, we argue that education about scholarly metrics and their limitations is of utmost importance. Online videos, which are often more efficient, effective and engaging than text, are a promising format to teach researchers and research administrators about metrics. This analysis of short videos about the h-index on YouTube demonstrated that, while many presenters make use of practices to balance cognitive load, they often lack critical discussions about the shortcomings of the indicator. We did not find any videos through our search that fulfilled the aim of our project, of producing high quality, thoughtfully designed, videos discussing the inherent problems of the h-index and its widespread use. We therefore argue that in order to make researchers and research administrators metrics literate, videos need to be produced by involving experts on content as well as online education and video production.

## Acknowledgments

**Disclaimer.** The views expressed in this presentation are those of the presenter and do not necessarily reflect the official policy or position of the Uniformed Services University of the Health Sciences, the Department of Defense, or the U.S. Government.

## Author Contributions

**Conceptualization:** Lauren A. Maggio, Alyssa Jeffrey, Stefanie Haustein.

**Data curation:** Lauren A. Maggio, Alyssa Jeffrey, Stefanie Haustein, Anita Samuel.

**Formal analysis:** Lauren A. Maggio, Alyssa Jeffrey, Stefanie Haustein, Anita Samuel.

**Funding acquisition:** Alyssa Jeffrey, Stefanie Haustein.

**Investigation:** Lauren A. Maggio, Alyssa Jeffrey, Stefanie Haustein, Anita Samuel.

**Methodology:** Lauren A. Maggio, Alyssa Jeffrey, Stefanie Haustein, Anita Samuel.

**Supervision:** Lauren A. Maggio, Stefanie Haustein.

**Writing – original draft:** Lauren A. Maggio, Alyssa Jeffrey, Stefanie Haustein, Anita Samuel.

**Writing – review & editing:** Lauren A. Maggio, Alyssa Jeffrey, Stefanie Haustein, Anita Samuel.

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
