## [Decision Letter · Decision Letter 0]

25 Feb 2022

PONE-D-22-02394Becoming metrics literate: An analysis of brief videos that teach about the h-indexPLOS ONE

Dear Dr. Maggio,

Thank you for submitting your manuscript to PLOS ONE. After careful consideration, we feel that it has merit but does not fully meet PLOS ONE’s publication criteria as it currently stands. Therefore, we invite you to submit a revised version of the manuscript that addresses the points raised during the review process. Please submit your revised manuscript by Apr 11 2022 11:59PM. If you will need more time than this to complete your revisions, please reply to this message or contact the journal office at plosone@plos.org. Please include the following items when submitting your revised manuscript:A rebuttal letter that responds to each point raised by the academic editor and reviewer(s). You should upload this letter as a separate file labeled 'Response to Reviewers'.A marked-up copy of your manuscript that highlights changes made to the original version. You should upload this as a separate file labeled 'Revised Manuscript with Track Changes'.An unmarked version of your revised paper without tracked changes. You should upload this as a separate file labeled 'Manuscript'.

We look forward to receiving your revised manuscript.

Kind regards,

Lutz Bornmann

Academic Editor

PLOS ONE

Journal Requirements:

3. Thank you for stating the following in your manuscript:

“Funding was provided by the Social Sciences and Humanities Research Council of Canada (SSHRC) Insight Grant #435-2021-0108 "Metrics Literacies: Improving the understanding and appropriate use of scholarly metrics in academia””

Please note that funding information should not appear in other areas of your manuscript. We will only publish funding information present in the Funding Statement section of the online submission form.

“SH and Aj were supported by funds from the 

Social Sciences and Humanities Research Council of Canada (SSHRC) Insight Grant #435-2021-0108 "Metrics Literacies: Improving the understanding and appropriate use of scholarly metrics in academia” (https://www.sshrc-crsh.gc.ca/)

Reviewers' comments:

Reviewer's Responses to Questions

**Comments to the Author**

1. Is the manuscript technically sound, and do the data support the conclusions?

Reviewer #1: Yes

Reviewer #2: Yes

2. Has the statistical analysis been performed appropriately and rigorously? 

Reviewer #1: N/A

Reviewer #2: N/A

3. Have the authors made all data underlying the findings in their manuscript fully available?

Reviewer #1: Yes

Reviewer #2: Yes

4. Is the manuscript presented in an intelligible fashion and written in standard English?

Reviewer #1: Yes

Reviewer #2: Yes

5. Review Comments to the Author

Reviewer #1: I highly appreciate this new approach to study information dispersion/education, in particular in the field of bibliometrics.

Some remarks though

p. 3 the authors’ definition of the h-index does not seem right. They say that a researcher with an h-index of 15 has published at least 15 papers which have received at least 15 citations each. An author having published at least 17 papers which have received at least 17 citations also meets this requirement. I think the authors should add that h is the largest number meeting their requirement.

p.4 among the problems with the h-index they should mention (already here) the dependence on the database, especially the difference between a Google h-index and a WoS h-index.

p.11 Although difficult I think that the authors should mention factual errors in these videos, or make a remark when they include statements which are clearly without any nuance or unethical, such as “My professor says...“. Also the statement that publishing in languages other than English is bad for one’s h-index, is besides the point. Publishing in other languages leads to less citations, and in this way influences the h-index.

p.15 A similar remark is valid when talking about gender bias. The h-index is not gender biased as such; citations are.

Some minor remarks

p.2 abstract. Line (-2). I would say a minority (32%) was of professional quality.

p. 6 line 3. I think that the word ‘thus’ should be removed.

p. 8 video 2. Wrong date

A final remark. The authors introduce the notion of “becoming metrics literate” (fine), but although mentioned, they do not state the difference with the term “metric-wise”. In my understanding, becoming metric-wise means becoming metric literate AND being able to use it to one’s advantage. I also note that the term “metric-wise” was discussed in the article mentioned by the authors, but actually introduced in (2015) “Metric-wiseness. JASIST, 66(11), 2389-2389.

Reviewer #2: This interesting article studies how videos on YouTbe explain and discuss the h-index. After a period in which the emtrics community was mostly oriented inwards (e.g., towards developing more sophisticated indicators), it is a welcome change to see increasing attention to the use of metrics by researchers, practitioners, and administrators. The present paper fits into the latter paradigm.

Overall, I think this paper is very well done, with sufficient attention to various aspects that make a video more or less suitable to learn about the h-index and its limitations. Similar studies could be done on other indicators and platforms, and I expect that that will happen in the coming years. The paper is clearly structured and well-written. I have only some fairly minor suggestions for the authors.

The paper presents solid arguments in favor of studying videos in particular. However, I'd like to point out that there are more options beyond mere text on the one hand and video on the other. For instance, the infographic developed by CWTS (https://leidenmadtrics.nl/articles/halt-the-h-index) presents some dreawbacks of the h-index in a static yet visual way.

Some additional information on the process that led to the final 31 videos would be appropriate. It is not clear to me if these were the first 31 videos presented by YouTube/Google, or if they constitute a truly random sample out of all 274. Table 1 shows that they represent a mixture of older and more recent videos, as well as more and less popular ones (which suggests some kind of randomization) but some more details shoud be provided. Can you also describe your findings after reviewing the last 7 videos? Did those yield any new insights or were they all 'more of the same'?

I would change the definition of the h-index on p. 3 to read that it is THE LARGEST NUMBER h such that there are h number of papers with at least h citations. (Strictly speaking, someone with h-index 15 also has 10 papers with 10 or more citations but in fact they have even more).

It would be helpful if the authors could provide some recommendations for future videos on the h-index or other bibliometric indicators: which formats work well? What are issues to be aware of?

6. PLOS authors have the option to publish the peer review history of their article (what does this mean?). If published, this will include your full peer review and any attached files.

Reviewer #1: No

Reviewer #2: No

---

## [Author Response · Author response to Decision Letter 0]

25 Mar 2022

We have uploaded our response to reviewers in our cover letter and as a separate document.

---

## [Decision Letter · Decision Letter 1]

25 Apr 2022

Becoming metrics literate: An analysis of brief videos that teach about the h-index

PONE-D-22-02394R1

Dear Dr. Maggio,

We’re pleased to inform you that your manuscript has been judged scientifically suitable for publication and will be formally accepted for publication once it meets all outstanding technical requirements.

Kind regards,

Lutz Bornmann

Academic Editor

PLOS ONE

Additional Editor Comments (optional):

Reviewers' comments:

Reviewer's Responses to Questions

**Comments to the Author**

1. If the authors have adequately addressed your comments raised in a previous round of review and you feel that this manuscript is now acceptable for publication, you may indicate that here to bypass the “Comments to the Author” section, enter your conflict of interest statement in the “Confidential to Editor” section, and submit your "Accept" recommendation.

Reviewer #1: All comments have been addressed

Reviewer #2: All comments have been addressed

2. Is the manuscript technically sound, and do the data support the conclusions?

Reviewer #1: (No Response)

Reviewer #2: (No Response)

3. Has the statistical analysis been performed appropriately and rigorously? 

Reviewer #1: (No Response)

Reviewer #2: (No Response)

4. Have the authors made all data underlying the findings in their manuscript fully available?

Reviewer #1: (No Response)

Reviewer #2: (No Response)

5. Is the manuscript presented in an intelligible fashion and written in standard English?

Reviewer #1: (No Response)

Reviewer #2: (No Response)

6. Review Comments to the Author

Reviewer #1: (No Response)

Reviewer #2: (No Response)

7. PLOS authors have the option to publish the peer review history of their article (what does this mean?). If published, this will include your full peer review and any attached files.

Reviewer #1: No

Reviewer #2: No

---

## [Editor Report · Acceptance letter]

29 Apr 2022

PONE-D-22-02394R1 

Becoming metrics literate: An analysis of brief videos that teach about the h-index 

Dear Dr. Maggio:

I'm pleased to inform you that your manuscript has been deemed suitable for publication in PLOS ONE. Congratulations! Your manuscript is now with our production department. 

Kind regards, 

on behalf of

Dr. Lutz Bornmann 

Academic Editor

PLOS ONE